# Coconut Mesocarp-Based Lignocellulosic Waste as a Substrate for Cellulase Production from High Promising Multienzyme-Producing *Bacillus amyloliquefaciens* FW2 without Pretreatments

**DOI:** 10.3390/microorganisms10020327

**Published:** 2022-01-31

**Authors:** Van Hong Thi Pham, Jaisoo Kim, Jeahong Shim, Soonwoong Chang, Woojin Chung

**Affiliations:** 1Department of Environmental Energy Engineering, Graduate School of Kyonggi University, Suwon 16227, Korea; vanhtpham@gmail.com; 2Department of Life Science, College of Natural Science of Kyonggi University, Suwon 16227, Korea; jkimtamu@kyonggi.ac.kr; 3Soil and Fertilizer Management Division, Rural Development Administration, National Institute of Agricultural Science, Wanju 54875, Korea; jaysoil@korea.kr; 4Department of Environmental Energy Engineering, College of Creative Engineering of Kyonggi University, Suwon 16227, Korea

**Keywords:** coconut-mesocarp degradation, organic waste degrading bacteria, multiple enzyme-producing bacteria, extremophiles, cellulase-producing bacteria

## Abstract

Facing the crucial issue of high cost in cellulase production from commercial celluloses, inexpensive lignocellulosic materials from agricultural wastes have been attractive. Therefore, several studies have focused on increasing the efficiency of cellulase production by potential microorganisms capable of secreting a high and diversified amount of enzymes using agricultural waste as valuable substrates. Especially, extremophilic bacteria play an important role in biorefinery due to their high value catalytic enzymes that are active even under harsh environmental conditions. Therefore, in this study, we aim to investigate the ability to produce cellulase from coconut-mesocarp of the potential bacterial strain FW2 that was isolated from kitchen food waste in South Korea. This strain was tolerant in a wide range of temperature (−6–75 °C, pH range (4.5–12)) and at high salt concentration up to 35% NaCl. The molecular weight of the purified cellulase produced from strain FW2 was estimated to be 55 kDa. Optimal conditions for the enzyme activity using commercial substrates were found to be 40–50 °C, pH 7.0–7.5, and 0–10% NaCl observed in 920 U/mL of CMCase, 1300 U/mL of Avicelase, and 150 U/mL of FPase. It was achieved in 650 U/mL, 720 U/mL, and 140 U/mL of CMCase, Avicelase, and FPase using coconut-mesocarp, respectively. The results revealed that enzyme production by strain FW2 may have significant commercial values for industry, argo-waste treatment, and other potential applications.

## 1. Introduction

The crisis of energy, the combustion of petroleum-based fossil fuels, and the rapid increase in agricultural municipal cellulosic waste have warned us about the picture of living in the future. These issues have shifted global efforts to explore and utilize renewable resources for the production of green energy and eco-environmental waste treatment strategies. One of the most valuable sources is lignocellulosic biomass, which has been identified as a great potential for bioconversion to value-added bioproduct from lignocellulose fermentation [1,2,3]. Cellulase is the second most important enzyme, only behind amylase due to its environmentally friendly and economical biofuels development [4,5,6,7,8,9]. Since the 1960s, cellulase has been used increasingly in food, paper, pulp, textile industries, and pharmaceutical industries. However, there are limited studies that exploited the hydrolysis process of agricultural residues of lignocellulosic materials to produce high-value products with low cost [10,11,12]. Moreover, the problem with the biorefining process is that it is costly due to a lack of biocatalysts that are capable of withstanding a variety of environmental stressors such as low/high temperature, acidic/alkaline pH, high salinity, and expression of multi-enzyme complexes. The development of cellulase production from extremophilic bacterial candidates has attracted microbiologists due to their high active research area and their greater enzyme yield than fungi [13,14]. Since bacterial cellulase showed significant difference in stability, catalytic potential, cellulose degradation efficiency, and achieving tremendous benefit if biomass utilization, the investigation of new resources for isolating desirable catalytic potential cellulase has been attracting more attention [15,16].

The main members of cellulase include endoglucanase, exoglucanase or cellobiohydrolase, and β-glucosidase [17]. *Bacillus* sp., *Clostridium* sp., *Cellulomonas*, *Thermomonospora, Ruminococcus*, and *Bacteroides* are the well-known cellulase-producing bacterial species which have been isolated from various sources [18,19,20]. Among them, numerous members from the *Bacillus* genus are promising enzyme-producing candidates due to their capacity to produce and secrete large quantities of extracellular enzymes [21,22,23,24].

Moreover, cellulose synthesis and recycling accounted for a larger proportion of the carbon cycle, approximately 80% of the biosphere, and over 90% of marine environments at a lower temperature of 5 °C [25]. The cellulose degradation period was prolonged at both low and high temperatures because of its complex structure. The complete degradation of cellulose requires the activity of diverse enzymes, such as cellulases [26]. Therefore, to accelerate and enhance the cellulose degradation process, different cellulose-degrading bacteria forming a variety of low temperature habitats have recently been of interest for isolation [27].

Therefore, psychrozymes produced from indigenous psychrophilic microorganisms have been considered as excellent biocatalysts owing to their high catalytic activity at low temperature. It eliminates the need for a heating process to save energy and to become a cost-effective alternative in the industrial application.

*Bacillus amyloliquefaciens* DL-3, a hydrolyzing rice hull bacterial strain was isolated, identified, and first characterized the produced cellulase for the utilization of cellulosic biomass by Lee and co-authors [18]. Ye et al., optimized the fermentation conditions and properties of cellulase-producing bacterium *B. amyloliquefaciens* S1 for high efficiency production and other green products for geese products [28]. In another study, *B. amyloliquefaciens*-ASK11 was explored for the ability of cellulase production under high chromium stress [29].

Since large amount of coconut residues continue to increase, numerous previous studies provided the possibility to address this problem by investigating the bio-ethanol production from coconut mesocarp-based substrates [30,31,32]. Moreover, coconut mesocarp was found to be the natural renewable raw bioresource of carboxymethyl cellulose that was earlier considered as a better carbon source and a major factor responsible for maximum cellulase production [14,33]. Therefore, Dey et al. firstly studied the cellulase production from such highly abundant lignocellulosic bio-waste material by cellulase produced from a fungal strain *Trichoderma reesei* [34]. However, in this study, we continued to screen and characterize the cellulase enzymes produced from a bacterial strain *B. amyloliquefaciens* FW2 that was isolated and identified from the previous study [24]. This strain showed a high ability to degrade the coconut mesocarp. The optimal growth and enzyme production conditions of bacterial strain were also investigated.

## 2. Materials and Methods

### 2.1. Isolation of Cellulose—Degrading Bacteria

Following isolation steps from the previous study, the kitchen food waste collected from the Jowon Industry in South Korea was used as the source for the functional bacterial isolation [14,24]. Five grams of food waste was added to 50 mL of distilled water to make an original isolation solution. One milliliter of sample was then added to the culture medium including (g/mL): carboxymethylcellulose (CMC) 10; MgSO_4_, 0.024, NaCl 10, K_2_HPO_4_ 0.3 and 10 mL of trace element mixture. The medium was adjusted to a pH of 7.0 at 25 °C. A vitamin solution of 10 mL and 10 mL of autoclaved soil extract (ASE) were added as the final step after autoclaving for 20 min (121 °C, 103 kPa). ASE was prepared by adding 100 g of soil into 1000 mL of distilled water and adjusted to pH 7 before autoclaving following Van Pham and Kim [35]. The vitamin solution contained (g/L): citric acid 0.02, folic acid 0.01, riboflavin 0.025, and para-amiobenzoic acid 0.01. The study targets to isolate the functional bacterial candidates that can be able to grow under extreme conditions. Therefore, these samples were incubated at different temperatures of −6 °C, 10 °C, 30 °C, 35 °C, 50 °C, 65 °C, 75 °C and 80 °C on a rotary shaker at 200 rpm (1 d interval for each temperature change). The bacterial culture in the medium (1 mL) was transferred to a fresh medium prepared as above and incubated at 30 °C for 5 d. Well-separated colonies were sub-cultured to obtain pure cultures, and subsequently examined for enzyme production and waste degradation ability in the subsequent steps [14].

### 2.2. Coconut Mesocarp Preparation

The coconut mesocarp was extracted from coconut fruit without liquid albumen using a stainless-steel knife and cut into smaller size. The raw material was dried at 105 °C in a forced-air oven to reduce the moisture content until it was less than 5% (*w*/*w*). Mesocarp material was then ground into powder, sieved with a 212-μm mesh, and stored inplastic zipper bags at 4 °C for future use.

### 2.3. Taxonomic Identification and Sequence Analysis

The bacterial strain FW2 was identified using the 16S rRNA gene by PCR amplification. Genomic DNA from the strain was extracted according to the manufacturer’s instructions using the InstaGene Matrix kit (Bio-Rad, Seoul, Korea). Following Frank et al., amplification of the 16S rRNA gene was then performed by PCR using primers (27F and 1492R) [36]. A multiscreen filter plate (Millipore Corp, Bedford, MA, USA) was used to purify the PCR products which were then sequenced using primers 518F (5′-CCA GCA GCC GCG GTA ATA CG-3′) and 800R (5′-TAC CAG GGT ATC TAA TCC-3′) with a PRISM BigDye Terminator v3.1 Cycle Sequencing Kit (Applied Biosystems, Foster City, CA, USA). This process was conducted at 95 °C for 5 min and then cooled on ice for 5 min and analyzed using an ABI Prism 3730XL DNA analyzer (Applied Biosystems, Foster City, CA, USA). Finally, the nearly full-length 16S rRNA sequence was assembled using SeqMan software (DNASTAR Inc., Madison, WI, USA). Sequence similarity was determined by comparison with the sequence available in the gen bank database using the EZBioCloud server [37].

Following the previous study, the FASTA sequences of related strains with strain FW2 obtained from the GenBank database were used in order to construct a phylogenetic tree [24]. The MEGA 7 program was used to align sequences and reconstruct the phylogenetic trees [38]. The best fit model used in this study for neighbor-joining (NJ) analysis was a Tamura 2-parameter model with gamma-distributed rates plus invariant site based on the minimum Bayesian information 140 criterion value (gamma parameter = 0.6 in this study) [39]. The reliability of the 141 phylogenetic trees was estimated by bootstrap values of 1500 replications [40].

### 2.4. Screening of Cellulase Production

To screen the extracellular enzyme production, bacterial strains were incubated in the basal medium with 1% KH_2_PO_4_, 0.25% Na_2_HPO_4_, 1% NaCl, 0.2% (NH_4_)_2_SO_4_, 0.005% MgSO_4_·7H_2_O, 0.005% CaCl_2_ added 1% of carboxymethylcellulose (CMC) 5 g or microcrystalline cellulose 5 g, and Yeast Extract 5 g. The growth and enzyme generation of strain FW2 were tested first at −6 °C and 80 °C following the previous study [14,24].

After 72 h of incubation, the plate was flooded with 1% iodine solution, and the clear zone around the colony indicated cellulose degradation by the produced bacterial enzyme.

### 2.5. Optimization of Enzyme Production

#### 2.5.1. Optimization of Physio-Chemical Parameters

The enzyme production process was examined with different experimental conditions such as various pH (4–12.5), temperature (−6–80 °C), and NaCl (0–35%). Different pH was maintained using appropriate buffers: 0.1 M citrate buffer (pH 4–5), 0.2 M phosphate (pH 6–8) and phosphate–NaOH buffer (pH 8–12.5).

#### 2.5.2. Effect of Carbon and Nitrogen Sources on Enzyme Production

Fermentation medium for strain FW2 was prepared in supplement of 1% (*w*/*v*) of each commercial cellulose substrate, including glucose, maltose, starch, dextrose, and lactose as various carbon sources. Other trials were added 1% (*w*/*v*) ammonium nitrate, potassium nitrate, yeast extract, casein, skim milk, and peptone as different nitrogen sources. The bacterial inoculated cultures were incubated at optimal temperature of 45 °C.

#### 2.5.3. Enzyme Production Using Coconut Mesocarp Powder

Cellulase yield of strain FW2 was examined using a different concentration of coconut mesocarp powder ranging from 1% to 10% (1% interval, *w*/*v*) in a flask containing 1 L of base medium including 1% KH_2_PO_4_, 0.25% Na_2_HPO_4_, 1% NaCl, 0.2% (NH_4_)_2_SO_4_, 0.005% MgSO_4_·7H_2_O, 0.005% CaCl_2_, and 5 g of yeast extract as nitrogen source. The culture was incubated in a shaking incubator and maintained at 45 °C, 150 rpm and 10% of bacterial culture (*v*/*v*) during 7 days. The culture was then filtered and centrifuged at 10,000 rpm for 5 min at 4 °C. The produced enzyme in solution was collected and used for the enzyme assay. All experiments were carried out in triplicates.

#### 2.5.4. Effect of Effects of Ions on Cellulase Activity

The effect of various metal ions on cellulase activity was determined by the presence of additives to 0.5 mL of crude enzyme preparation. The additives (NaCl, KCl, MgCl_2_, FeSO_4_, CaCl_2,_ and ZnCl_2_) were used at final concentration of 5.0 mM. The reaction mixtures were incubated with the additives for 60 min at 37 °C at pH 5.0. Residual cellulase activity was measured by DNS (3,5-dinitrosalicylic acid) method [41]. The relative (%) value was calculated and compared with the control tube without any metal ion.

### 2.6. Cellulase Activity Assay

The bacterial strain was cultured in individual enrichment medium for cellulase fermentation, it contained (g/L): NaCl 5, peptone 5, yeast 10, CMC 5 (for CMCase), or microcrystalline cellulose (for Avicelase), and KH_2_PO_4_ 1 at 45 °C and 150 rpm for 5 d.

The cellulase assay was performed by extracting the supernatant of the bacterial culture after centrifugation at 10,000 rpm at 4 °C. The reaction mixture contained 0.5 mL of different crude enzyme dilutions and 0.5 mL of 1% CMC as a substrate (in 0.1 M citrate buffer, pH 4.8). The mixture was incubated at 50 °C for 30 min, and the reaction was terminated by adding 3 mL of DNS solution, and the solution was boiled for exactly 5 min for color development. All samples were then cooled rapidly. The reduction in sugar was estimated spectrophotometrically at 540 nm following the method of Miller [41]. One unit of cellulase activity was defined as the amount of enzyme required to liberate 1 μmol of reducing sugars (measured as 2 mg of glucose) per milliliter per minute under assay conditions.

Assay of filter paper activity (FPase) was estimated using gravimetric determination. The bacterial culture media was composed of (g/L): KH_2_PO_4_, 0.5; MgSO_4_, 0.25; gelatin 2, Whatman filter paper 50 mg/20 mL/L of distilled water containing the target bacterial inocula. After 3 d of incubation, the cultures were centrifuged at 6000 rpm for 15 min at 4 °C. The collected pellets were used to estimate the constant weight after drying by comparing them with the trial without bacterial inocula. All experiments were performed in triplicates.

### 2.7. Purification of Cellulase Enzymes

The crude cellulase enzyme was extracted from the culture under optimal conditions and then precipitated overnight using ammonium sulphate at concentration of 40% and 80% saturation. The pellets were collected by centrifugation at 10,000 rpm, 4 °C for 10 min, and suspended in 50 mM phosphate buffer (pH 7) and dialyzed overnight. The enzyme sample was conducted with an anion exchange chromatography using a DEAE-Cellulose column equilibrated with phosphate buffer pH 8.0.

### 2.8. Molecular Weight Determination

The purity and molecular weight of the sample at each step was examined by Sodium Dodecyl Sulphate Poly Acrylamide Gel Electrophoresis (SDS-PAGE) using the method described by Laemmli on Mini Protean Tetra System (Bio Rad Laboratories, Inc., Hercules, CA, USA) [42]. The enzyme was separated on the 12 separating gel and 4% stacking gel. Electrophoresis was carried out for about 30 min at 200 V, and the protein bands were visualized with Comassie Brilliant Blue R-250 staining.

## 3. Results

### 3.1. Identification of Cellulase-Producing Bacterial Strain

The phylogenetic bacterial strain FW2 in this study was identified by comparing the sequence of the amplified 16S rRNA gene against sequences deposited in the GenBank database with Accession No. MW652625. Based on the 16S rRNA sequences data, the strain FW2 had the highest homology with *Bacillus amyloliquefaciens* DSM 7^T^ (99.86%) and pairwise similarity with other members shown in Table 1 and Figure 1 [24].

### 3.2. Screening the Cellulase Production of Strain FW2

Enzyme activities were confirmed by clear zones after staining with iodine solution. Clearance > 1.0 cm was considered significant. Cellulases were observed even at pH 4.5 and 35% NaCl, and were weakly active at pH 4 and 40% NaCl. Cellulose and CMC degradation were weak at 80 °C. Optimal enzyme production profiles were obtained at pH 7–8, NaCl 0–10%, and temperature 40–45 °C (Figure 2). The result show that cellulases are strongly active in the pH range of 5–10.

### 3.3. Effect of Culture Conditions on Cellulase Activity of Strain FW2

#### 3.3.1. Effect of Physio-Chemical Parameters and Enzyme Production Stability

The largest sizes of the halo zone for cellulose and CMC degradation were 53 mm, and 48 mm, respectively, at 45 °C for 4 days (Figure 2c,d).

CMCase enzyme activity was determined from the clearance zone detected by iodine staining during the screening test on agar plates supplemented with 1% CMC, filter paper was added for total cellulase (FPase), and crystal cellulose was used for Avicelase at a wide range of parameters: −6 °C to 75 °C and weak at 80 °C, pH 4.5–12, and NaCl concentration at 0–35%. CMCase production increased from 35 °C and reached a halo zone peak of 48 mm measured at 40–45 °C, while Avicelase reached the highest production observed with 53 mm in diameter at 45 °C after 4 d of incubation, respectively.

In the same pattern of CMCase, FPase and Avicelase enzymes showed the highest production at 54 h of incubation time with 920 U/mL, 150 U/mL, and 1300 U/mL at 40–45 °C and pH 7–7.5, respectively, before a gradual decrease at 60 °C (Figure 3 and Figure 4).

Especially, compared with other enzymes, this enzyme stability was high, retaining 90 at 45 °C and 71% at 75 °C after 12 h of incubation. Moreover, under acidic condition of pH 4.5 and alkaline pH 12, it was active with 59% and 35% after 12 h, respectively (Figure 5 and Figure 6). CMCase was stable at −6 °C with 74% after 2 h and retained only 15% after 12 h of incubation. However, its stability was observed to be 90% at 45 °C, 66% at 75 °C, 65% at pH 4.5, and 51% at pH 12 (Figure 6).

#### 3.3.2. Effect of Carbon and Nitrogen Sources on Enzyme Production 

In the enzyme assays, the optimal fermentation parameters were determined. As shown in Figure 7a, glucose, dextrose, and maltose exhibited an effect on CMCase production at 795, 815, and 802 U/mL, respectively. It was revealed that lactose was the most effective, producing 910 U/mL and 1250 U/mL, while starch was a less suitable carbon source for cellulase production with only 770 U/mL and 670 U/mL of CMCase and Avicelase, respectively. The medium supplemented with yeast exhibited the highest enzyme production of 920 U/mL and 1300 U/mL, indicating that yeast was the best nitrogen source for CMCase and Avicelase production by strain FW2, respectively. These were followed by tryptone (890 U/mL), casein (875 U/mL), skim milk, and potassium nitrate in the same amount (860 U/mL), while minimum effect was exhibited by ammonium chloride (790 U/mL) for CMCase and Avicelase, yield was exhibited by potassium nitrate observed at 770 U/mL (Figure 7b).

#### 3.3.3. Effect of Coconut Mesocarp Substrate on Cellulase Production at Various Concentrations

Figure 8 shows the effect of coconut mesocarp bio-base substrates on cellulase production. CMCase and Avicelase yielded the highest amounts observed at 650 U/mL and 720 U/mL at 4% of the substrate, respectively, whereas maximum FPase production was found at 3% of the substrate achieved in 140 U/mL.

#### 3.3.4. Effect of Metal Ions on Cellulase Activity

The result in Table 2 illustrates that the addition of Mg^2+^ and Ca2^+^ enhanced all types of cellulase enzymes. The presence of Mg^2+^ accelerated CMCase yield up to 138.5%, followed by FPase and Avicelase accounting for 130.5% and 120%, respectively. While Na^+^ had no effect on CMCase, both Na^+^ and K^+^ had a negligible effect on both FPase and Avicelase, accounting for around 95–99%. Fe^2+^ and Zn^2+^ had a negative effect on Avicelase by reducing it between 11% and 17%, on CMCase by between 4% and 15%, and on FPase by between 2% and 21%, respectively.

### 3.4. Determination of Molecular Weight of Extracted Cellulase

The homogenous enzyme preparation was obtained by SDS-PAGE analysis shown in Figure 9. Molecular weight mass of purified cellulase was estimated to be 55 kDa.

## 4. Discussion

Following the previous study, *Bacillus* sp. FW2 was able to grow and produce enzyme under harsh conditions such as pH 4.5, pH 12, concentrations of NaCl up to 35% and withstand 75 °C. However, the related strains of strain FW2 grow within a limited temperature ranging from 15 to 45 °C [24,43]. In the other study, some halophilic bacterial strains were observed to yield cellulase at pH 6–12, temperature range of 30–90 °C, and up to 20% NaCl [44]. In this study, cellulase activity in the range of pH 5–10 by strain FW2 was consistent with the results of previous studies [45,46]. The optimal pH 7–7.5 for the bacterial strain FW2 was similar to the strain of *B. amyloliquefaciens* [18].

To date, there have been few studies on enzyme production at low temperatures. The cellulase produced by some psychrophilic microorganisms demonstrated optimal activity in the acidic to neutral pH range 4.5–7.0 [47,48]. However, other bacteria belonging to the genera *Paenibacillus*, *Pseudoalteromonas*, and *Shewanella* showed strong cellulase activity at neutral to alkaline pH [49,50]. Recently, thermostable enzyme production from bacteria has attracted considerable attention owing to its application in a wide range of fields. These thermo-enzymes are stable at high temperatures and active under other extreme conditions, such as varying pH values and salt concentrations [51,52,53,54]. However, these bacterial strains exhibited only one type of enzyme under thermophilic conditions. On the other hand, *B. amyloliquefaciens* DL-3, investigated in the previous study, showed high thermal stability at broad temperatures ranging from 40 to 80 °C [18].

Bioconversion of valuable products from lignocellulosic biomass requires combined pretreatment processes and catalytic degradation of that substrate. In such a way, plant dry materials in the form of lignocellulosic wastes were used to produce value-added products such as bio-ethanol, xylitol, and carboxylic acids [55,56,57,58]. However, the cellulase produced from various types of lignocellulosic materials was less observed using bacteria in previous studies in comparison with this study (Table 3).

There is no study on coconut mesocarp application of cellulase production by a fungal strain *Trichoderma reesei* besides the study of Dey et al. [34]. This study was the second in the research of bioconversion from coconut mesocarp, one of the prominent agricultural wastes in tropical countries using the high potential cellulose degrading *Bacillus* strain FW2 without any pretreatment. Moreover, coconut mesocarp is considered as a potential feed stock for production of value-added products including bioethanol, due to the presence of high content of cellulose (43.4%) and hemicellulose (19.9%) [66]. However, to achieve the high production efficiency of bio-products, the various pretreatments were carried out for breaking down the crystalline structure of lignocelluloses materials using physio-chemical techniques, such as liquid hot water [67] and sodium hydroxide treatment [10]. In the recent study, production of cellulase under optimized nutritional conditions was arrived at through the use of a response surface technique in Design Expert Software (version 8.0.4) [34].

Metal ions can form complexes in association with proteins and other molecules related to enzymes. They may act as donors or acceptors of the electron as structural regulators [68]. Mg^2+^ and Ca^2+^ were considered as the additives to enhance cellulase activity in many previous studies. The stimulatory effect of Mg^2+^ and Ca^2+^ on cellulase was also reported by Yoon et al., and Bakare et al. [69,70]. Fe^2+^ reduced FPase activity by 33.4% [71]. The negative effect of Fe^2+^ on cellulase was also reported by Yin et al. [72]. The production of cellulase was enhanced by the addition of NaCl and MgSO_4_ in another study [73].

The molecular mass of the purified cellulase produced from bacterial strain FW2 was estimated to be 55 kDa, which was found in the cellulase of *Acinetobacter junii* GAC 16.2, while it was 53 kDa of *B. amyloliquefaciens* DL-3 [18,74].

## 5. Conclusions

The isolate *Bacillus amyloliquefaciens* FW2 was investigated as a promising cellulase producer using various carbon and nitrogen sources. Moreover, this bacterial strain was able to degrade the coconut mesocarp that contains a high amount of lignin and hemicelluloses without pretreatment. Therefore, strain FW2 may be considered in the future as an effective degrader for more types of agricultural wastes. This study continues to explore the functional bacterial candidates that allow cost-effective production of cellulase and an ideal for a clean environment and biomass waste management. Further research should be carried out to optimize the fermentation conditions for the production of cellulase from this bacterial strain using coconut mesocarp in scale-up before application in a larger scale.

## Figures and Tables

**Figure 1 microorganisms-10-00327-f001:**
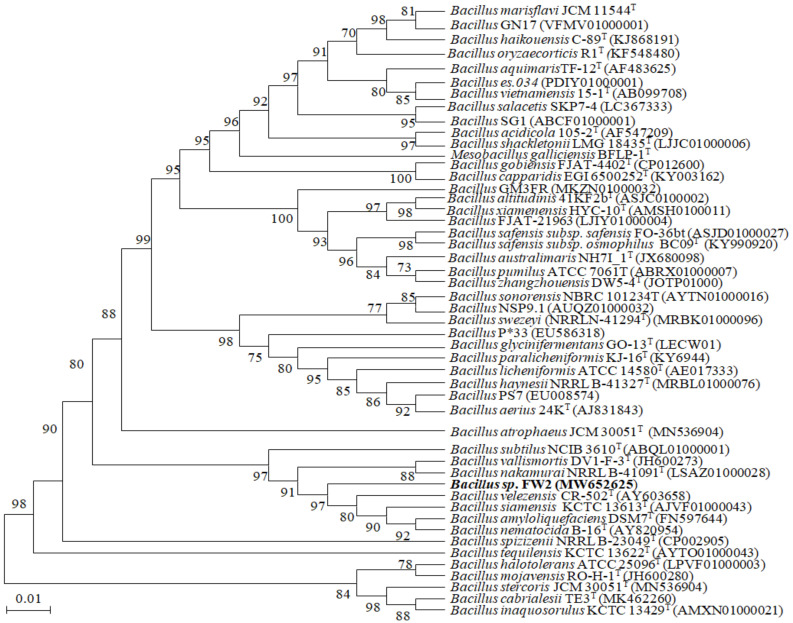
A detailed tree that displays the phylogeny of 16S rRNA gene sequences of strain FW2 and related Bacillus genus members. The tree was constructed using the neighbor-joining method. Bootstrapping was carried out with 1500 replicates.

**Figure 2 microorganisms-10-00327-f002:**
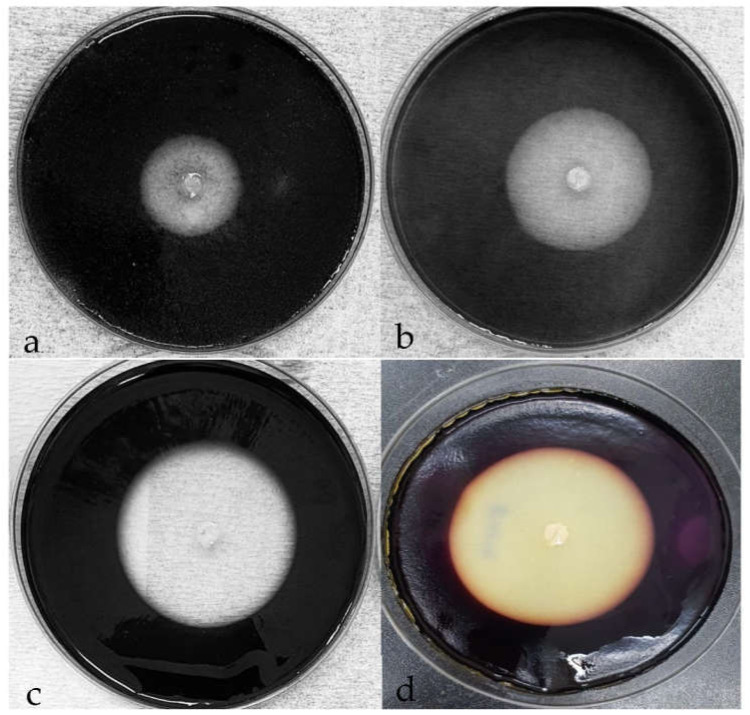
The growth and cellulose degradation of strain FW2 at 45 °C for 4 days: (**a**) at pH 12; (**b**) in the medium supplemented with 35% NaCl. The enzyme production under optimal conditions in 4-day incubation (pH 7–7.5; 45 °C): (**c**) cellulose degradation by CMCase; (**d**) cellulose degradation by Avicelase.

**Figure 3 microorganisms-10-00327-f003:**
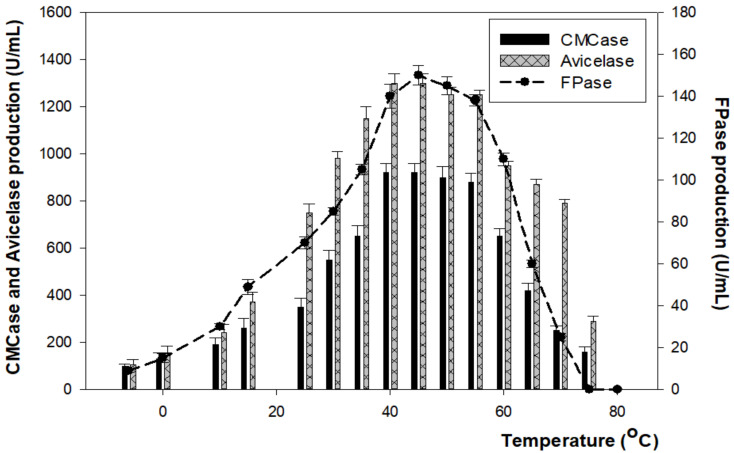
The effect of temperature on enzyme production. Samples were incubated for 4 days at pH 7–7.5.

**Figure 4 microorganisms-10-00327-f004:**
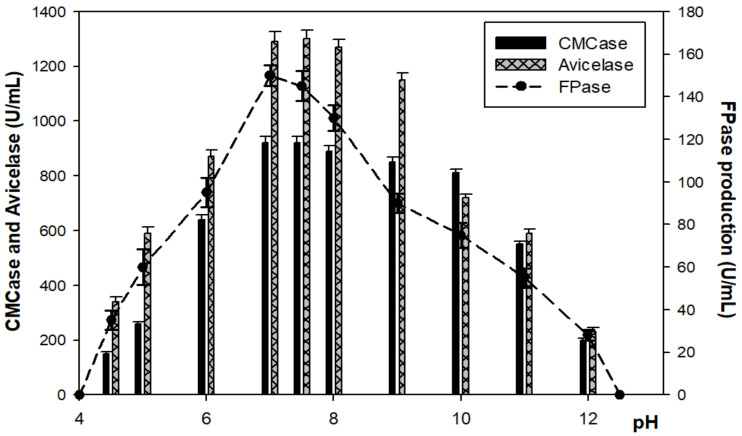
The effect of pH on enzyme production. Samples were incubated for 4 days at 45 °C.

**Figure 5 microorganisms-10-00327-f005:**
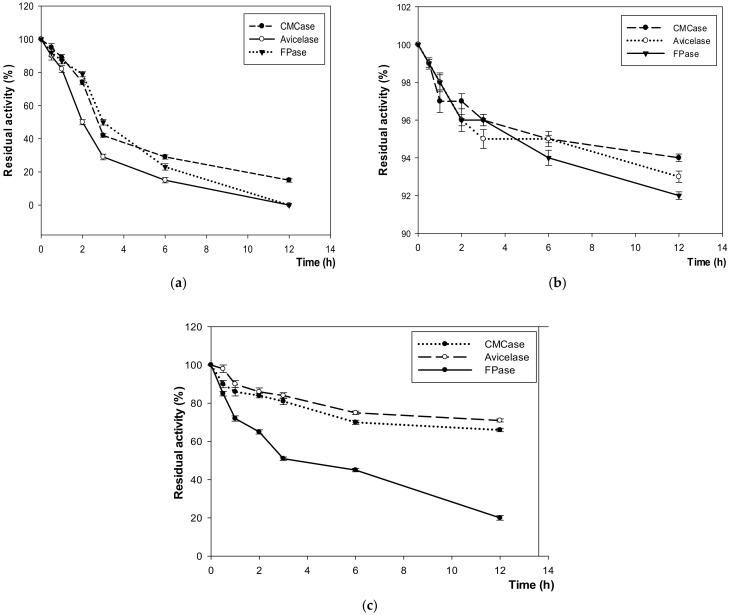
The stability of enzymes at different extremes and optimal temperatures: (**a**) −6 °C; (**b**) 45 °C and (**c**) 75 °C.

**Figure 6 microorganisms-10-00327-f006:**
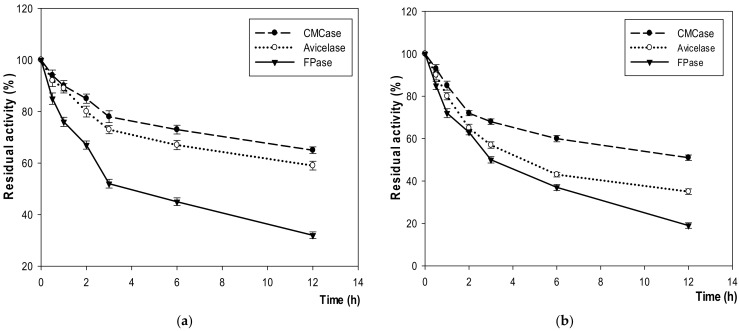
The stability of enzymes at (**a**) pH 4.5 and (**b**) pH 12.

**Figure 7 microorganisms-10-00327-f007:**
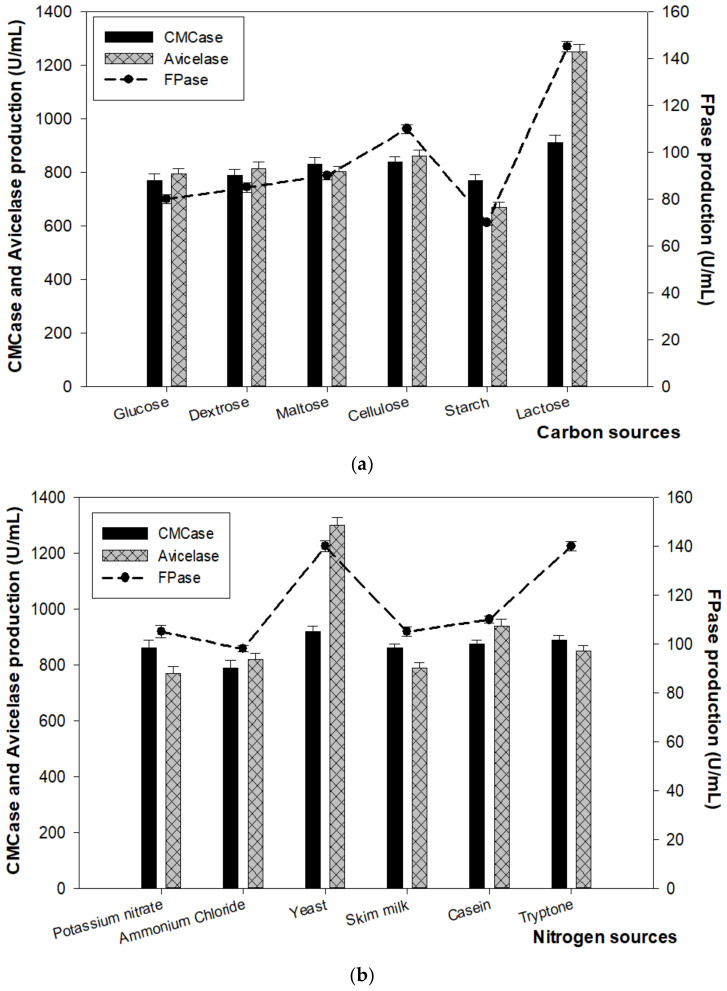
The effect of carbon sources (**a**) and nitrogen sources (**b**) on enzyme production. Samples were incubated at 45 °C and pH 7–7.5 for 4 days.

**Figure 8 microorganisms-10-00327-f008:**
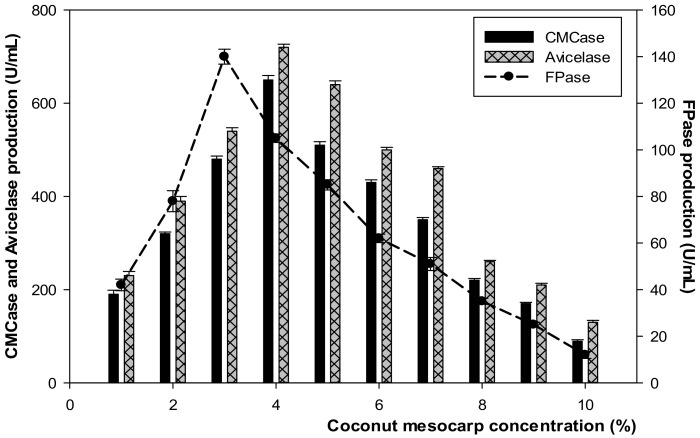
Production of cellulase enzymes at different concentrations of the coconut mesocarp without pretreatment. Samples were incubated at 45 °C, pH 7–7.5.

**Figure 9 microorganisms-10-00327-f009:**
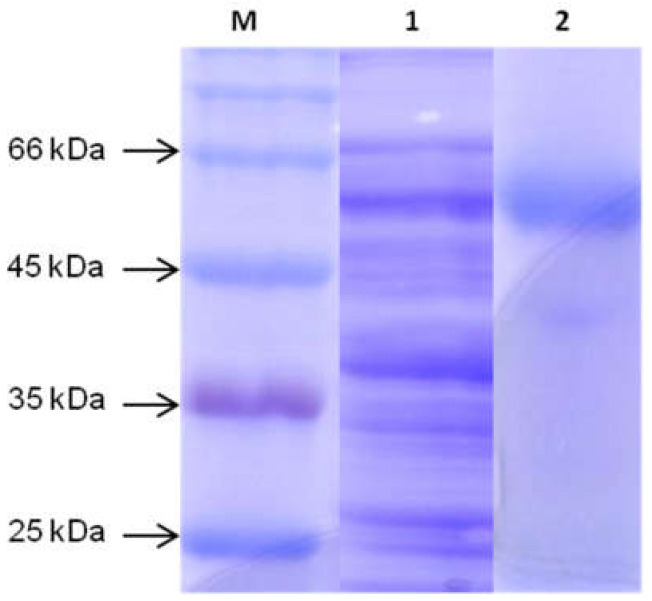
SDS-Polyacrylamide gel electrophoresis of the purified cellulase produced from B. amyloliquefaciences FW2. (**M**) protein molecular makers; (**1**) supernatant culture broth; (**2**) purified enzyme.

**Table 1 microorganisms-10-00327-t001:** Related bacterial strains with *B. amyloliquefaciens* FW2 based on the similarity of the 16S rRNA sequences.

Strain	Similarity (%)	Different Nucleotide/Comparison
*Bacillus amyloliquefaciens* DSM7^T^	99.86	2/1472
*Bacillus siamensis* KCTC 13613^T^	99.86	2/1472
*Bacillus velezensis* CR-502^T^	99.86	2/1403
*Bacillus nematocida* B-16^T^	99.73	4/1470
*Bacillus subtilis* NCIB 3610^T^	99.59	6/1472
*Bacillus nakamurai* NRRL B-41091^T^	99.59	6/1472
*Bacillus cabrialesii* TE3^T^	99.52	7/1472
*Bacillus inaquosorum* KCTC 13429^T^	99.52	7/1472
*Bacillus stercoris* JCM 30051^T^	99.52	7/1472
*Bacillus vallismortis*	99.46	8/1472

**Table 2 microorganisms-10-00327-t002:** Effect of metal ions on cellulase activity of strain FW2.

Metal Ions	Residual Activity (%)		
	FPase	CMCase	Avicelase
Control	100	100	100
Na^+^	99	100	98
K^+^	97	99	95
Mg^2+^	130.5	138.5	120
Fe^2+^	98	96	89
Ca^2+^	115.8	122.5	108
Zn^2+^	79	85	83

**Table 3 microorganisms-10-00327-t003:** Cellulase produced from various types of agricultural waste using bacterial strains in recent studies.

Substrate	Bacterial Strains	Cellulase Production (U/mL)	Reference
Potato peel	*Bacillus subtilis* K-18	CMCase (3.5)	[59]
Alkali-pretreated corn cob	*Bacillus* sp. BS-5	CMCase (9.6), FPase (1.4)	[60]
Rice straw residues	*Bacillus cereus* RSI6	CMCase (0.36)	[61]
Sugarcane bagasse	*Paenibacillus polymyxa* ND25	CMCase (0.49)	[62]
Wheat bran	*Bacillus* sp. AO	CMCase (5.9), FPase (0.97)	[63]
Algal biomasses	*Bacillus* sp. TPF-1	CMCase (9.12)	[64]
Corn husk	*Sphingobacterium* sp. ksn-11	CMCase (3.55)	[65]
Coconut mesocarp	*Bacillus sp.* FW2	CMCase (650), Avicelase (720)	This study

## Data Availability

The data used to support the findings of this study are available from the corresponding author upon request.

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
