# Peer review of "Coconut Mesocarp-Based Lignocellulosic Waste as a Substrate for Cellulase Production from High Promising Multienzyme-Producing Bacillus amyloliquefaciens FW2 without Pretreatments"

_microorganisms, 2022, doi:10.3390/microorganisms10020327_

Round 1

Reviewer 1 Report

1) Many stylistic, spelling and lexical errors

2) No details about the purification of the enzyme, still it would be interesting since the homogenious enzyme was obtained in only ONE chromatographic step

3) Is it the ONLY enzyme of enzymatic coctail that reveal the cellulase activity? Data not shown. 

4) What was the approximate degree of coconut mesocarp conversion by Bacillus  sp. FW2? 

Author Response

Dear Reviewer, 

We would like to thank you very much for your valuable time and helpful comments in our manuscript to make it more perfect before it is able to be accepted for publishing in this journal.

Please find the attachment for our response.

All the best,

Woojin Chung

Reviewer 2 Report

The topic of the study fits within the scope of the journal, but there are still few substantial remaining questions/comments listed below in detail:

But first of all: The percentage of the cited literature in the range of “2014 or older” is >60%. From this perspective, it would be necessary to cite the current literature in order to justify more thoroughly the novelty of the present work and/or further substantiate the proposed progress. The update of the references would have impact on the sections “introduction/state of the art” and  “results and discussion”!

Apropos references: please check all of them carefully since e.g. [11] can’t get accessed (what means “in progress” exactly?) and the title of [12] is wrong.

In addition, it would be helpful to include a table with already published figures comparing the results here obtained.

Could you please elaborate the need/targets (lines 102-104) of the low temperature(s)…?

Obviously (if I haven't missed it somewhere else...) some experimental details are missing in section 2.5. Could you please describe the cultivation equipment including the scale/quantities and specifically for 2.5.3 the "base medium"? The latter (although stated in the title coconut mesocarp) is too concise and only explained by two lines 163/164.

All the graphs show standard deviations, but just for the section 2.6 triplicates (3-fold analysis of the same sample?) are mentioned. If I understand it correctly the cellulase activity assay is just the “analytical method” to determine the performance mainly for the experiments of section 2.5. Could you please comment on this?

Please harmonize the scaling of the y-axes in figures 4, 5 and 8 for better comparability.

Author Response

(The authors gave the same response as above.)

Round 2

Reviewer 2 Report

The authors tried to provide additional explanations and changes according to my initial recommendations. But the changed/added parts compared to the original submission are not really comprehensive in all single questions. First of all again references – please try to improve/update the citations (still >40% “2014 or older”) together with the “introduction” and “results & discussion”, respectively. There are still gaps answering the points 4 (for any industrial application those extreme/harsh conditions wouldn’t be of interest I guess), point 5 (cultivation equipment, scale/quantities) and point 7 (the scaling of the y-axes in figures 4, 5 and - now - 7 is still not harmonized).

Author Response

Dear Reviewer,

First of all, we would like to thank you again for your insightful comments and valuable suggestions on our manuscript in Round 2. We have revised the manuscript following each comment. The change was highlighted in yellow. We hope that our response of questions listed as below will make you satisfied.

Comments and Suggestions for Authors

The authors tried to provide additional explanations and changes according to my initial recommendations. But the changed/added parts compared to the original submission are not really comprehensive in all single questions.

  1. First of all again references – please try to improve/update the citations (still >40% “2014 or older”) together with the “introduction” and “results & discussion”, respectively.

Response: We would like thank you again for this useful comment. We updated the more recent studies instead of the old ones that were highlighted in yellow in the whole text and reference list.

  1. There are still gaps answering the points 4 (for any industrial application those extreme/harsh conditions wouldn’t be of interest I guess).

Response:  We sincerely apologize for the previous explanation that may be not clear to you. Actually, in our current research areas, this cellulase producing bacterial strain used in this study was isolated from the previous studies. In which organic compound degrading bacterial candidates were the target of our research. Especially, our food waste composting was carried out during the winter that room temperature was low around 10-16 ºC. However, the preferable temperatures for composting should be over 25 ºC.  As one of the most important purposes for saving energy from heating the composting room, we attended the psychophilic bacterial strains. Therefore, low temperature tolerant bacterial strain FW2 was found in that situation. After that, the effect of temperature to its growth and its bio-product production in all next studies will be tested from the low to high temperatures.

We hope that our explanation will make you clear and satisfied.

  1. point 5 (cultivation equipment, scale/quantities)

 Response: We would like thank you very much for reminding us again about this point. We have rewritten the section 2.5.3 to make it more concise and detail. Please check it in yellow highlight.

  1. point 7 (the scaling of the y-axes in figures 4, 5 and - now - 7 is still not harmonized).

Response: We sincerely apology for this mistake. We have harmonized the Figure 4.5 and 7.

Finally, we would like thank you for your valuable time and your helpful comments to make our manuscript better before it can be accepted for the publication.

Best regards,

Woojin Chung
